# How Do Network Embeddedness and Environmental Awareness Affect Farmers' Participation in Improving Rural Human Settlements?

**Jinhua Xie** [1,2]**, Gangqiao Yang** [1,]*****, Ge Wang** [1,3] **and Wei Xia** [1]

1   College of Public Administration, Huazhong Agricultural University, 1 Shizishan Street,
    Wuhan 430070, China; xiejinhua01@webmail.hzau.edu.cn (J.X.); ge.wang@mail.hzau.edu.cn (G.W.);
    xw123@webmail.hzau.edu.cn (W.X.)
2   Department of City and Regional Planning, the University of North Carolina at Chapel Hill,
    Chapel Hill, NC 27599, USA
3   Antai College of Economics and Management, Shanghai Jiao Tong University, 1954 Huashan Road,
    Shanghai 200030, China
*   Correspondence: ygq@mail.hzau.edu.cn; Tel.: +86-1397-166-9174

**Abstract:** Based on social embeddedness theory, this paper aims to explore the influence mechanism of network embeddedness and environmental awareness on farmers' participation in improving rural human settlements (IRHS). This research applies the Logit model and the Bootstrap method, using survey data from 495 farmers in Hubei Province, China. The results show that: (1) relational embeddedness has a significant negative impact on the centralized treatment of farmers' domestic sewage, implying that strengthening the relationship between farmers and households helps to provide them with centralized treatment for domestic sewage; (2) environmental awareness has a significant positive impact on the centralized treatment of farmers' domestic sewage, implying that the enhancement of farmers' environmental awareness increases the promotion centralized treatment for domestic sewage; and (3) structural embeddedness can further affects farmers' environmental awareness and then affects their participation in the centralized treatment of domestic sewage, implying that environmental awareness has a mediating effect between structural embeddedness and the centralized treatment of farmers' domestic sewage. Overall, it is necessary not only to encourage the establishment of extension and discussion networks for farmers (relational embeddedness) to participate in IRHS but also to improve environmental education for farmers, especially by increasing their access to environmental knowledge and information (environmental awareness in mountainous areas, and, finally to support farmers. The relationship between the members and the village cadres (structural embeddedness) can further improve farmers' awareness of participation in IRHS to better guide them in the centralized treatment of domestic waste and domestic sewage.

**Keywords:** network embeddedness; environmental awareness; improving rural human settlements (IRHS); farmers' participation behavior

## 1. Introduction

Currently, China is experiencing unprecedented rapid urbanization, but many major issues prevail in rural development, leading the government to vigorously promote rural revitalization. With rapid population growth and economic and urban development, domestic waste management has become a pervasive global pressure [1]. Developing countries also experience rural domestic waste management as a serious challenge [2], while there is no distinction between domestic waste management in urban and rural areas in developed countries [3]. Although China's agricultural economy continues to grow and farmers' incomes continue to rise, littering and the indiscriminate discharge of sewage are still a common picture in many rural areas. In the past 40 years, with the rapid development of the rural economy and the change of lifestyle, the quantity and variety

of rural household garbage have increased rapidly. In 2017, China's total domestic waste reached 213 million tons and is expected to grow by up to 300 million tons in 2020 [4]. According to the statistical data from the Development Prospects and Investment Forecast Analysis Report of China's Rural Waste Treatment Industry from 2018 to 2023, the output of rural household garbage in China was about 180 million tons in 2017, with a per capita daily output of 0.8 kg. Currently, there are at least 70 million tons of domestic waste that have not been treated [5]. In many villages, domestic waste is piled up in a disorganized way, burned in the open air, surrounded by garbage, and blocked by garbage, which not only breeds germs and spreads disease but also pollutes the land, groundwater, and surface water [6]. Rural domestic waste has become one of the main pollution sources in the rural environment. Large amounts of household garbage exacerbate environmental harm by posing an imminent threat to safety. While the main waste disposal measures are still mixed landfill (53.0%) and mixed incineration (43.8%) [7], the promotion of waste classification and the reduction in the waste volume are fundamental to solving the waste problem, as these actions can reduce environmental pollution, save land, promote the recycling of resources, and improve public value.

At present, some untreated domestic waste and sewage are directly discharged into rivers, lakes, and other surface water bodies in certain underdeveloped rural areas in China, which considerably affects the living environment and the water bodies [8]. Due to their high operational management requirements, high costs, and water quality standards [9], there are some challenges to using integrated treatment processes in these relatively dispersed areas (such as townships). The results of the implementation of numerous rural governance measures in the countryside have attracted increasing attention from policymakers. The implementation effect of these initiatives mainly depends on whether and to what extent local farmers are actively encouraged to participate in improving rural human settlements (IRHS) [10]. Rural households are the direct beneficiaries of and important participants in IRHS [11], and their attitude toward IRHS is crucial to the policy formulation and financing of IRHS [10]. Therefore, research on farmers' participation in IRHS and its influencing mechanism is urgently needed. Furthermore, this study discusses its formation mechanism from the perspective of social embeddedness theory. Therefore, improving rural human settlements is one of the important tasks within the rural revitalization strategy.

Recently, with the great improvement of material living standards, rural residents' demand for a better environment has increasingly grown. The availability of public goods and local infrastructure and the effective governance of rural cooperatives are related to the success or failure of the construction of a beautiful village. Many sociologists assume that environmental issues are ultimately caused by irrational individual environmental behaviors [12]. This understanding of ecological and environmental problems emphasizes the role of individuals in coping with the current ecological crisis. Consequently, there is a growing public concern about environmental sustainability issues and the impact of human behavior on natural ecosystems [13,14], which also elevates participation in IRHS to a high priority in public debate [15].

The concept of rural human settlements is derived from the concept of human settlement environments, and rural human settlements are a kind of non-material organism that is related to agricultural production and farmer reproduction [16]. According to Wu's definition, rural human settlements include the natural system, human system, residential system, social system, and support system; IRHS is the main area of focus through which the support system can improve important aspects of farmers' living standards, including rural garbage control, toilet waste control, domestic sewage control, upgrades in the appearance of villages, village planning, village management, etc. [17]. IRHS relates to the supply of public goods and is also concerned with the characteristics of private goods because the government attends to IRHS while the farmers contribute to the improvement of the environment around their houses. Farmers' participation in IRHS is guided by the government to promote the appearance of villages and, taking the periphery of rural households as

the model, to attend to the centralized treatment of domestic waste and domestic sewage, human and animal waste treatment, environmental greening, debris management, and other activities [10,18]. Although IRHS covers a range of features, depending on the rural reality, this study mainly focuses on two aspects: the centralized treatment of domestic waste and domestic sewage.

IRHS can mitigate environmental damage through direct and indirect means and improve the environment individually or collectively [19]. It is influenced by internal and external factors [20–22]. Without knowledge of the environment, consciously caring about it or adopting an environmentally conscious attitude is not possible [23]. Varela-Candamio et al. [24] argue that governments should ascribe importance to public environmental education, improve public environmental knowledge, and promote more environmentally friendly behavior. According to Hungerford and Volk [25], environmental education is different from other forms of general education because consciousness does not necessarily promote behavior. However, Frick et al. [26] and Bartiaux [27] found no correlation between environmental knowledge and behavior. Recently, the issues of solid waste discharge [28–30], domestic sewage discharge [31–35], and rural household toilet reform [36–38] have received much attention. Farmers' participation in agroecological production [39–42], agricultural land rehabilitation [43,44], and environmental management [45–48] has been discussed, and it provides a useful reference for this paper.

China's rural community is a typical "acquaintances society", with the characteristics of differential patterns, in which the behavior of rural households is not only the result of rational individual decision-making but also of conformity with group decision-making [49–51]. Individuals can influence those around them to participate in environmental protection through learning, interaction, and reciprocity within their relationship networks [52]. These rural social networks perform multiple functions, such as the spread of information and social learning, which can effectively promote farmers' participation in IRHS [53–55]. Another key factor affecting farmers' participation in IRHS is environmental awareness. The level of environmental awareness directly affects whether farmers contribute to IRHS and is also influenced by social networks because communication between farmers can effectively improve their environmental awareness.

Through our literature review, we found that, although the existing research in different dimensions discusses farmers' participation in IRHS, further expansion in the following three aspects remains to be addressed. Firstly, few studies analyze the environmental impact mechanism of farmers' IRHS and did not reveal the influence of regional differences. Secondly, although some scholars have discussed the influence of farmers' participation in IRHS from the perspective of social networks or environmental awareness, most have not incorporated these three aspects into one research framework. Few studies have considered the possible mediating role of environmental awareness between them. Furthermore, the mechanism through which network embeddedness affects farmers' participation in IRHS through their environmental awareness is still unclear. Thirdly, most existing studies use Logit or Probit models for analysis, which are not conducive to identifying the critical path affecting farmers' participation in IRHS, nor the direct and indirect effects of various factors. Thus, to propose a research framework and to further study the mechanism behind farmers' participation in IRHS from the perspective of social networks and environmental awareness, and then to better guide and promote their participation in IRHS, this study used questionnaire survey data from farmers in some counties and cities in central China to construct an analytical framework of the impact mechanism of network embeddedness and environmental awareness on farmers' participation in IRHS. By using the Logit model and Bootstrap method to explore its mechanism, and by guiding farmers to actively participate in IRHS more effectively, it will be possible to carry out comprehensive and well-organized land management and ecological restoration in the future. This study focused on assessing how network embeddedness and environmental awareness shape farmers' participation in IRHS and formulated the following research questions (RQs).

RQ1: What are the challenges faced by typical villages in improving rural human settlements during the operational stage?

RQ2: What is the scope of application of different governance models for improving rural human settlements in typical villages?

The results of this research could inform strategies to reduce environmental damage through interventions (e.g., fostering environmental attitudes and emotions) influence individual participation in IRHS.

The remainder of this paper is organized as follows. The second part constitutes the theoretical analysis and research hypotheses. Section 3 presents the data sources and research methods, including the population and sample studies, measurement, and analysis strategies. The Section 4 introduces the results of this empirical study, including the model fitting test, structural results, mediating effect analysis, and multi-group analysis. Section 5 discusses the findings. Finally, the conclusion and problems for further research are presented.

## 2. Theoretical Analysis

Although the improvement of rural human settlement environments has been put on the agenda and achieved some positive results, the overall progress level is still not high, and there are widespread problems of imbalance and insufficiency [10]. Until 2017, the domestic garbage treatment rate in rural areas of China reached 72.99%, of which the domestic waste treatment rate was only 23.62% and the rural domestic sewage treatment rate was 17.19% [10]. To fundamentally improve the rural human environment, the Chinese government has put forward the rural human environment improvement action plan and issued a series of documents, including the rural revitalization strategy [56] and the three-year action plan for rural human environment improvement [57]. In January 2019, the Central Agricultural Office, the Ministry of Agriculture and Rural Affairs, and other 18 departments jointly developed the "rural habitat environment improvement village cleaning action program". This program calls for mobilizing farmers to participate in rural environmental remediation and focus on solving rural environmental problems [58]. These improvements in the rural human environment can provide farmers with a clean and tidy village environment, complete infrastructure, and sound public services [11,59]. These can effectively improve farmers' well-being and promote the construction of rural ecological civilization [60].

Based on social embeddedness theory and combined with current research on the effect of network embeddedness [52,61] and environmental awareness [23] on farmers' participation in IRHS, this study constructs the following research framework: network embeddedness, environmental awareness, farmer participation in IRHS" (Figure 1). Social embeddedness theory divides network embeddedness into two types: relational embeddedness and structural embeddedness [62–64]. Specifically, relational embeddedness refers to the relational network formed by embedding rural households' behaviors in the surrounding villagers [65]. Structural embeddedness refers to the network structure embedded in rural households and their position in the network [66]. Environmental awareness refers to farmers' basic awareness of the ecological environment and their understanding and grasp of relevant knowledge based on ecosystem services, which is the basis for farmers to construct ecological values [67]. Referring to an existing study [28], farmers' participation in IRHS was represented in this paper by their participation in the treatment of domestic waste and sewage. Therefore, the main research topics were as follows: (1) the direct effect of network embeddedness on farmers' participation in IRHS; (2) the direct influence of environmental awareness on farmers' participation in IRHS; (3) analysis of the mediating effect of environmental awareness between network embeddedness and farmers' participation in IRHS.

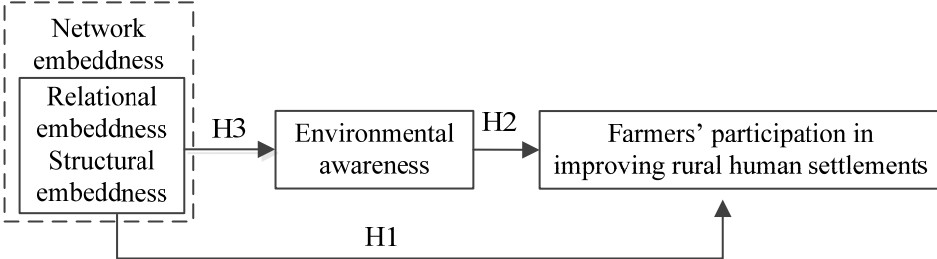

**Figure 1.** The mechanism influencing network embeddedness and environmental awareness on farmers' participation in IRHS.

*2.1. Direct Impact of Network Embeddedness on Farmers' Participation in IRHS*

Network embeddedness originates from Polanyi's idea of "the social embedding of the economy" [68], while Granovetter extended the social embeddedness theory and developed an important argument in favor of "the embedding of economic behavior in social structure" [62], which argues that human economic activities are conducted based on a certain network of relationships and have been embedded in the network structure. The economic behaviors of rural households are usually to a certain extent, restricted by the structure of the social relations in which they are located. They are not only independent but also "embedded" in a certain network of relations. While network embeddedness is mainly composed of relational embeddedness and structural embeddedness [62–64], relational embeddedness mainly includes relationship strength and relationship quality [65], and structural embeddedness mainly includes network size, network density, and network location [66]. The quality and strength of relationships between farmers have a direct impact on the knowledge that farmers can acquire. Although explicit knowledge can be obtained through observation or imitation, tacit knowledge can only be obtained through higher-quality relationships [64,69]. Good interpersonal relationships and emotional closeness among relatives and friends can help farmers to share tacit knowledge. This further strengthens trust and cohesion among farmers, promotes the establishment of a mutually beneficial cooperation mechanism among farmers [52], and further improves farmers' participation in IRHS. Therefore, this paper proposes the following research hypothesis:

**Hypothesis 1a.** *Relational embeddedness has a positive impact on farmers' participation in IRHS.*

In addition, the social network in which farmers are embedded is conducive to information sharing and communication among farmers, and the formation of a villager relationship network assists in promoting farmers to engage in more in-depth, extensive, and effective information exchange [66]. Farmers' participation in IRHS is easily affected by the behaviors of surrounding farmers, especially those who play an important role in the network of villagers [70]. The complex network formed among farmers can provide them with more important information, promote the dissemination and sharing of knowledge related to rural ecological environment governance, and guide them to participate in IRHS in practice. Therefore, this paper proposes the following research hypothesis:

**Hypothesis 1b.** *Structural embeddedness has a positive impact on farmers' participation in IRHS.*

*2.2. Direct Impact of Environmental Awareness on Farmers' Participation in IRHS*

Environmental awareness is a subjective feeling and awareness of environmental conditions and environment-related issues and refers to the way individuals think about environmental issues and related political actions [67]. The relationship between environmental awareness and environmental protection behavior has been widely discussed in academic circles. According to some studies, no significant correlation exists between environmental awareness and individual pro-environmental behavior [71], which might

be attributed to individual environmental awareness not being related to actual pollution. Environmental pollution may not be sufficient to stimulate individual environmental protection behavior [72]. However, most scholars believe that the more knowledge individuals possess about the potential risks and threats that may arise from environmental problems, the more attention they pay to the solution of environmental problems and the improvement of environmental quality, and the more likely they are to adopt more responsible and positive environmental protection behaviors [73–77]. In other words, individual environmental awareness may have a certain influence on a person's environmental behavior [23]. Furthermore, individual environmental awareness first affects their environmental attitude, then affects their sense of responsibility, and finally affects their environmental behavior [73]. Similarly, the deeper farmers' understanding of the rural ecological environment, the higher their awareness, and the more likely they are to participate in the governance of rural human settlements [78]. Therefore, this paper proposes the following research hypothesis:

**Hypothesis 2.** *Environmental awareness has a positive impact on farmers' participation in IRHS.*

*2.3. Mediating Effect Analysis of Environmental Awareness*

The different areas in which individuals live and the different elements in their social networks lead to differences in individual awareness of the environment. Network embeddedness has many functions (e.g., risk avoidance, information sharing) that affect individual environmental awareness. Similarly, the long-term formation of a variety of relationship networks among farmers is conducive to promoting information exchange and sharing, which can effectively reduce information asymmetry [61]. Through the efficient dissemination of environment-related knowledge and information, farmers' environmental awareness can be improved and their sense of responsibility enhanced, which will finally assist them in consciously participating in environmental activities (e.g., IRHS) [73,75,79,80]. In addition, the deeper farmers' understanding of IRHS, the more likely they are to contribute to IRHS [81]. In other words, network embeddedness first promotes farmers' environmental awareness and then affects their participation in IRHS. Therefore, this paper proposes the following research hypotheses H3a and H3b:

**Hypothesis 3a.** *Environmental awareness has a mediating effect on the influence of relational embeddedness on farmers' participation in IRHS.*

**Hypothesis 3b.** *Environmental awareness has a mediating effect on the influence of structural embeddedness on farmers' participation in IRHS.*

## 3. Research Area and Data Overview

The data used in this paper originate from the questionnaire survey conducted by the College of Public Administration of Huazhong Agricultural University in Wuhan City, Huangshi City, Jingmen City, and Tianmen City of Hubei Province in November 2019. To ensure data quality, the research team participated in detailed survey training and guidance for researchers. Based on the research plan, the survey process included the following elements. The poll was conducted according to the stratified random sampling method to extract Wuhan City, Huangshi City, Jingmen City, and Tianmen City as the study area. Further classification identified the Huangpi district in Wuhan City, Daye county-level city in Huangshi City, Jingshan county-level city, Zhongxiang county-level city in Jingmen City, and Tianmen City as samples areas. Three streets, five towns, and one township were randomly selected from these five sample areas. Next, according to the level of economic development of each town (or street, or township), three or four administrative villages were randomly selected in each sample town (or street, or township); the selected villages are all typical villages. Finally, 20–30 rural households were randomly selected in each sample administrative village as the survey subjects, and face-to-face interviews were

conducted in households or the field. Most of the respondents to the survey were heads of rural households or family decision-makers.

A total of 520 questionnaires were distributed in this survey, and 495 valid survey responses were obtained with an effective rate of 95.19%. Specifically, 89, 112, 215, and 79 valid questionnaires were collected in Wuhan City, Huangshi City, Jingmen City, and Tianmen City, respectively. The questionnaire mainly included the individual characteristics, family characteristics, social network, environmental awareness, and the interviewees' level of participation in IRHS. All the collected sample questionnaires were checked and reviewed by the research group and then collected to establish a sample database.

## 4. Analysis Method and Variable Setting

### 4.1. Analysis Method

Combined with the above theoretical analysis and existing research [23,52,67,81], and because the explained variables were dummy variables, this paper adopted the Logit model for empirical analysis with the following preliminarily equation:

$$Y = \beta_0 + \beta_1 NE + \beta_2 EC + \beta_3 Control + \varepsilon \tag{1}$$

Equation (1), $Y$ is the explained variable, namely, the farmers' participation in IRHS. $NE$ and $EC$ denote network embeddedness and environmental awareness, respectively. Additionally, based on existing research and considering data availability, the control variables are included in equation (1) as *Control*. $\beta_0$ is a constant term, $\beta_1$, $\beta_2$, and $\beta_3$ are the corresponding regression coefficients of the above explanatory variables, and $\varepsilon$ is a random disturbance term.

### 4.2. Variable Settings

According to the actual situation in the study area and existing research results [3,55], the corresponding variables were selected (Table 1). Specifically, the explained variable was whether farmers participated in IRHS, including "whether they participate in the centralized treatment of domestic waste" [3] and "whether they participate in the centralized treatment of domestic sewage" [55]. The core explanatory variables included network embeddedness and environmental awareness. Network embeddedness included relational embeddedness and structural embeddedness [64]. Relational embeddedness consisted of relationship strength and relationship quality [65]. Structural embeddedness consisted of network size, network density, and network location [66]. The control variables mainly included individual characteristics [9], household characteristics [82,83], village characteristics [84], regional social and economic characteristics [85], and other indicators. Additionally, the results of the collinearity test showed the absence of any significant multicollinearity among the variables [86,87], which met the requirements of econometric analysis.

**Table 1.** Variable values of surveyed households.

| Variable | Variable Meaning and Assignment | Mean Value | Standard Deviation |
|---|---|---|---|
| Explained variable | | | |
| Whether to participate in the centralized treatment of domestic waste ($Y_1$) | Yes = 1, no = 0. Farmers dumping waste in open spaces, roadsides, ditches, and rivers were considered as not participating in the centralized treatment of domestic waste, no = 0; when farmers threw living garbage into the cesspit, into the trash can (pool), or carried out the centralized classification of household garbage when removing this kind of garbage, or if the garbage was subjected to decomposition, such actions were regarded as participating in the centralized treatment of domestic waste, yes = 1. | 0.91 | 0.29 |
| Whether to participate in the centralized treatment of domestic sewage ($Y_2$) | Yes = 1, no = 0. Farmers passing the sewage through the drain into rivers, roadsides, ditches, yards, etc., or pouring it into the field via an infiltration pool were all regarded as not participating in the centralized treatment of domestic sewage, no = 0; farmers collecting and discharging domestic sewage through ta sewer, or collecting domestic sewage through the sewer and then purifying it, were regarded as the centralized treatment of domestic sewage, yes = 1. | 0.66 | 0.47 |
| Core explanatory variable | | | |
| Network embeddedness | | | |
| Relational embeddedness | | | |
| Relationship intensity | Do you often lend property (e.g., farm tools, machinery) to friends or neighbors? Yes = 1, no = 0 | 0.74 | 0.44 |
| Relationship quality | Do you trust your house to your neighbor when you go out? Yes = 1, no = 0 | 0.79 | 0.41 |
| Structural embeddedness | | | |
| Network size | Do you know many people in your local area? Very few = 1, few = 2, usual = 3, many = 4, very many = 5 | 3.43 | 0.85 |
| Network density | Your contact with relatives and family members. No contact = 1, occasional contact = 2, general = 3, more contact = 4, frequent contact = 5 | 3.72 | 0.80 |
| | Your contact with non-relatives of villagers and village cadres. No contact = 1, occasional contact = 2, general = 3, more contact = 4, frequent contact = 5 | 3.08 | 2.02 |
| Network location | How well you are respected by the local villagers. Very respectful = 1, somewhat respectful = 2, general = 3, somewhat disrespectful = 4, very disrespectful = 5 | 2.55 | 0.63 |
| Environment awareness | Do you care about the quality of the surrounding environment? Very unconcerned = 1, relatively unconcerned = 2, general = 3, relatively concerned = 4, very concerned = 5 | 3.87 | 1.51 |
| | What do you think of the surrounding ecological environment? Very poor = 1, relatively poor = 2, average = 3, fairly good = 4, very good = 5 | 3.67 | 0.83 |

**Table 1.** *Cont.*

| Variable | Variable Meaning and Assignment | Mean Value | Standard Deviation |
|---|---|---|---|
| Control variables | | | |
| Gender of the household head | Male = 1, female = 0 | 0.94 | 0.24 |
| Age of the household head | Under 35 = 1, 35-45 = 2, 45-55 = 3, 55-65 = 4, 65+ = 5 | 3.92 | 0.95 |
| Education level of the household head | Illiteracy = 1, primary school = 2, junior high school = 3, high school or technical secondary school = 4, junior college and above = 5 | 2.54 | 0.90 |
| Whether village cadres (leader) are in the family | Yes = 1, no = 0 | 0.07 | 0.26 |
| Whether family members are party members | Yes = 1, no = 0 | 0.08 | 0.26 |
| Contracted land area | The area is subject to the confirmation and certification of the second land contract management right (unit: mu) | 11.92 | 22.83 |
| Family size | The total population of rural households | 3.05 | 1.09 |
| Annual household income level | Total annual income of each labor force in rural households, unit: yuan. Less than 25,000 = 1, 25,000-50,000 = 2, 50,000-75,000 = 3, 75,000-100,000 = 4, more than 100,000 = 5 | 3.15 | 1.73 |
| The proportion of agricultural income | The proportion of agricultural income in the total income of rural households | 0.21 | 0.33 |
| Regional social and economic development level | According to each county area (city), the economic development level is divided. High = 1, low = 0 | 0.59 | 0.49 |
| Whether to carry out industrial integration | Whether the village has carried out industrial integration and the development of related industries and projects, yes = 1, no = 0 | 0.31 | 0.46 |

## 5. Empirical Analysis

*5.1. Descriptive Analysis*

The statistical results of the farmers' participation in IRHS are shown in Table 2.[1] Regarding the farmers' participation in the centralized treatment of domestic waste, in general, 450 households treated their garbage, accounting for 90.91% of the total sample. A total of 328 households, accounting for 66.26% of the total sample, participated in the centralized treatment of domestic sewage. To some extent, the high participation rate shows that Hubei Province, as a pilot province of overall domestic waste and the centralized treatment of domestic sewage at the county level, achieved good results in rural environmental treatment. There were some differences in the proportion of domestic waste and domestic sewage treated centrally in different regions. In Jingshan county-level city, Zhongxiang county-level city, Tianmen city, and Huangpi district, the proportion of domestic waste's centralized treatment by farmers was more than 90%, while in Daye county-level city the proportion was less than 90%. This indicates that the proportion of the centralized treatment of domestic waste was higher in the study area. This proportion increased in line with increasing levels of economic development, possibly because the requirements of farmers for environmental quality also increase with the level of economic development in their area. This stimulates them to perform the centralized treatment of domestic waste. The proportion of rural households that centrally treated their domestic sewage also displayed some differences. In Zhongxiang county-level city and Huangpi district, the proportion of rural households with centralized sewage treatment exceeded 70%, while in Jingshan county-level city, Tianmen city, and Daye county-level city, the proportion was less than 70%, indicating that the proportion of rural households in the research area using domestic sewage centralized treatment was higher. Similar to the centralized treatment of domestic waste and the reasons for participation, the proportion of centralized sewage treatment increased along with rising levels of economic development. However, it was difficult to obtain a convincing conclusion based only on the descriptive statistical analysis results [88], and quantitative analysis was needed to better analyze the mechanisms influencing farmers' participation in IRHS.

*5.2. Quantitative Analysis*

According to the theoretical analysis above, the quantitative analysis discussed in this paper was performed in the following two steps: firstly, benchmark regression was conducted to preliminarily test the influence of network embeddedness and environmental awareness on farmers' participation in IRHS; secondly, the stepwise regression method and the Bootstrap method were used to investigate the mediating effect of environmental awareness.

5.2.1. Test of the Influence Mechanism of Network Embeddedness and Environmental Awareness on Farmers' Participation in IRHS

(1) Preliminary test of core explanatory variables. Table 3 shows the regression results through the gradual introduction of explanatory variables. Only two variables embedded in the network were included in Model 1 and Model 2. The logarithmic probability function values of these two dependent variables of "rural household participating in the treatment of domestic waste" and "rural household participating in the treatment of rural domestic sewage" were -150 and -311.8, respectively, and the pseudo-$R^2$ values were 0.0054 and 0.0148. Models 3 and 4 introduced environmental awareness variables based on Model 1 and Model 2, respectively, and the logarithmic probability function value of the models increased to -149.5 and -290.6, and the pseudo-$R^2$ increased to 0.0088 and 0.0816, indicating that the explanatory power of the models was enhanced after the inclusion of the environmental awareness variables. By introducing control variables based on Model 3 and Model 4, the pseudo-$R^2$ values of the obtained Model 5 and Model 6 increased to 0.114 and 0.133, which further enhanced the explanatory power of the models. In conclusion, the estimation results of Model 1 to Model 6 showed that the significance and direction of the

core explanatory variables did not change significantly. This indicates that the estimation results were relatively robust. The following analysis is mainly based on the estimated results of Model 5 and Model 6.

The results of Model 5 in Table 3 indicate that the influences of network embeddedness on farmers' participation in IRHS varied greatly for each dimension. Specifically, each dimension of network embeddedness and environmental awareness did not exert any significant effect on farmers' participation in the centralized treatment of domestic waste. As a possible explanation, China is currently vigorously promoting the construction of an ecological civilization. Specifically, the government has introduced several policies and regulations to guide and encourage the protection of the ecological environment, invested in the construction of rural infrastructures, such as garbage disposal facilities, and regularly organized staff to pick up rubbish (pool). Village cadres (leaders) also inspect villages regularly and conduct additional irregular inspections. The implementation of these measures has resulted in the centralized treatment of domestic waste and sewage. Although some farmers still "litter", most farmers consciously perform the centralized treatment of domestic waste, as reflected by the descriptive statistical results. Furthermore, network embeddedness and environmental awareness had no significant effect on farmers' participation in the centralized treatment of domestic waste.

The results of Model 6 show that relational embeddedness had a significant effect on farmers' participation in the centralized treatment of domestic sewage (P-value is less than 5%), but the coefficient was negative, and the effect of structural embeddedness was close to the significance level of 10%. This indicates that relational embeddedness had a significant inhibiting effect on the treatment of domestic sewage, while structural embeddedness had a certain promoting effect. Therefore, hypothesis H1 was verified. As an explanation, maintaining a good interpersonal relationship with the surrounding farmers (the average relationship quality and relationship intensity were 0.74 and 0.79, respectively) helped to reduce the transaction cost of their information exchange and thus promoted homogeneity among their behaviors. The education level of rural households in the study area was generally not high (2.54, between elementary school and junior high school), and nearly 34% of farmers still did not participate in the centralized treatment of domestic sewage. This type of behavior involved a "demonstration effect", or, in psychological terms, a "broken window effect", which served to propagate the view that participation in domestic sewage centralized treatment is "bad" for the surrounding farmers. To some extent, however, structural embeddedness helped to encourage the farmers to participate in the centralized treatment of domestic sewage. In particular, close contact with relatives and family members, as well as with the kin and, especially, close relations between village officials, can stimulate farmers to take part in the centralized treatment of domestic sewage. Considering farmers and relatives, family members and villagers, and more closely linked village cadres, the constraints of family members and village cadres are more likely to influence farmers' behaviors. The constraints of family members are mainly reflected in the influence of influential members in the family. The constraints of village cadres are mainly reflected in the direct or indirect criticism of the non-environmentally friendly behaviors of rural households by village cadres [89]. Village cadres usually play a bridging role to connect the government and farmers and strike a balance between institutional supervision and farmers' trust. A good relationship between village cadres and farmers can promote farmers' participation in local village affairs [90]. Finally, network embeddedness can influence farmers to participate in the treatment of domestic sewage. The results of Model 6 also showed that environmental awareness had a significant positive effect on the treatment of domestic sewage. When farmers' environmental awareness is increased, their concern about the environmental quality of their surroundings increases, their willingness for and behavior toward environmental protection consciously improves, and they are more likely to centrally treat domestic sewage. In conclusion, hypothesis H2 was partially verified.

**Table 2.** Farmers' participation in IRHS.

| District | Number of Farmers in the Sample (Households) | Farmers Participating in Centralized Treatment of Household Garbage (Households) | Farmers Not Treating Household Garbage (Households) | Percentage of Domestic Waste Centralized Treatment (%) | Farmers Participating in Centralized Treatment of Domestic Sewage (Households) | Farmers Not Treating Domestic Sewage (Households) | Percentage of Rural Households with Domestic Sewage Centralized Treatment (%) |
|---|---|---|---|---|---|---|---|
| Daye city | 112 | 88 | 24 | 0.79 | 64 | 48 | 0.57 |
| Jingshan city | 80 | 75 | 5 | 0.94 | 52 | 28 | 0.65 |
| Zhongxiang city | 135 | 128 | 7 | 0.95 | 107 | 28 | 0.79 |
| Tianmen city | 79 | 75 | 4 | 0.95 | 42 | 37 | 0.53 |
| Huangpi district | 89 | 84 | 4 | 0.94 | 63 | 26 | 0.71 |
| Total | 495 | 450 | 45 | | 328 | 167 | |

**Table 3.** Logit regression results of farmers' participation in IRHS.

| Variable | Model 1 | Model 2 | Model 3 | Model 4 | Model 5 | Model 6 |
|---|---|---|---|---|---|---|
| | $Y_1$ | $Y_2$ | $Y_1$ | $Y_2$ | $Y_1$ | $Y_2$ |
| Relational embeddedness | 0.37 | −0.59 ** | 0.39 | −0.57 ** | −0.08 | −0.62 ** |
| | (0.38) | (0.27) | (0.38) | (0.28) | (0.43) | (0.30) |
| Structural embeddedness | 0.13 | 0.30 ** | 0.10 | 0.17 | 0.18 | 0.17 |
| | (0.19) | (0.13) | (0.19) | (0.14) | (0.25) | (0.14) |
| Environment awareness | | | 0.21 | 0.90 *** | 0.24 | 0.86 *** |
| | | | (0.21) | (0.15) | (0.22) | (0.16) |
| Control variables | Control | Control | Control | Control | Control | Control |
| Constant | 1.59 ** | 0.15 | 0.91 | −2.70 *** | 1.22 | −4.09 *** |
| | (0.67) | (0.46) | (0.94) | (0.67) | (1.67) | (1.05) |
| Number of observations | 495 | 495 | 495 | 495 | 495 | 495 |
| Log probability | −150 | −311.8 | −149.5 | −290.6 | −133.7 | −274.2 |
| Pseudo-$R^2$ | 0.005 | 0.015 | 0.009 | 0.082 | 0.114 | 0.133 |
| Chi$^2$ | 1.631 | 9.353 | 2.654 | 51.66 | 34.24 | 84.46 |

Note: The values in parentheses are standard errors; ***: $p < 0.01$, **: $p < 0.05$.

(2) The robustness test of the core explanatory variables. To test the robustness of the baseline regression results and to consider the differences between farmers in regions with different geomorphic types (e.g., plains and mountains), this paper divided the samples into plain samples and hilly-area samples, according to geomorphic types, for further analysis. According to Table 4, the results of Models 9, 11, and 5 were relatively consistent, and the results of Models 10, 12, and 6 were also relatively consistent, namely, network embeddedness and environmental awareness had significant effects on farmers' participation in IRHS. This indicated that the results of the benchmark regression were stable. In addition, when comparing the Logit model results of Model 5 and 6 (Table 3) with the OLS model results of Model 7 and 8 (Table 4), the estimation results of the Logit model and the OLS model were consistent in terms of the size, direction, and significance of the coefficients of the variables. These results further indicated that the benchmark regression results were robust.

**Table 4.** Regression results of the robustness test.

| Variable | Model 7 | Model 8 | Model 9 (Plain) | Model 10 (Plain) | Model 11 (Hilly Area) | Model 12 (Hilly Area) |
|---|---|---|---|---|---|---|
| | OLS | OLS | Logit | Logit | Logit | Logit |
| | $Y_1$ | $Y_2$ | $Y_1$ | $Y_2$ | $Y_1$ | $Y_2$ |
| Relational embeddedness | 0.00 | −0.10 * | −1.00 | −2.18 *** | 0.12 | 0.27 |
| (RE) | (0.03) | (0.05) | (1.11) | (0.65) | (0.53) | (0.39) |
| Structural embeddedness | 0.01 | 0.02 | −0.21 | 0.29 | 0.57 | 0.12 |
| (SE) | (0.01) | (0.02) | (0.41) | (0.23) | (0.37) | (0.14) |
| Environmental Awareness | 0.02 | 0.15 *** | 0.34 | 0.70 *** | 0.06 | 1.05 *** |
| (EC) | (0.02) | (0.03) | (0.39) | (0.23) | (0.30) | (0.23) |
| Control variables | Control | Control | Control | Control | Control | Control |
| Constant | 0.81 *** | −0.23 | 2.81 | −3.13 * | 0.16 | −4.93 *** |
| | (0.12) | (0.19) | (2.71) | (1.64) | (2.25) | (1.53) |
| Number of observations | 495 | 495 | 268 | 294 | 201 | 201 |
| Log-probability | - | - | −56.86 | −146.5 | −68.45 | −112.4 |
| $R^2$ | 0.07 | 0.15 | - | - | - | - |
| F | 2.596 | 6.069 | - | - | - | - |
| Adj-$R^2$ | 0.0433 | 0.126 | - | - | - | - |
| pseudo-$R^2$ | - | - | 0.0618 | 0.201 | 0.175 | 0.150 |
| $Chi^2$ | - | - | 7.491 | 73.91 | 28.98 | 39.68 |

Note: The values in parentheses are standard errors; ***: $p < 0.01$, *: $p < 0.1$.

### 5.2.2. Test of Mediating Effect of Environmental Awareness

(1) Preliminary test of mediating effect. The results of Model 6 showed that relational embeddedness had a direct negative impact, environmental awareness had a direct positive impact, and structural embeddedness had no direct impact on the treatment of domestic sewage. To further test whether network embeddedness indirectly affected the farmers' participation in IRHS through environmental awareness, it was necessary to estimate the impact of network embeddedness on farmers' environmental awareness. According to Table 5, structural embeddedness had a significant promoting effect on environmental awareness. These results indicated that environmental awareness had a complete mediating effect between structural embeddedness and farmers' treatment of domestic sewage, but the robustness of the mediating effect needed to be verified further. Furthermore, relational embeddedness only had a direct negative impact on the centralized treatment of domestic sewage.

**Table 5.** Impact of network embeddedness on farmers' environmental awareness.

| Variable | Environmental Awareness (EC) | |
|---|---|---|
| | Coefficient | Standard Error |
| Relational embeddedness (RE) | -0.04 | (0.09) |
| Structural embeddedness (SE) | 0.07 ** | (0.03) |
| Constant | 3.00 *** | (0.29) |
| Control variables | Control | |
| Number of observations | 495 | |
| $R^2$ | 0.07 | |
| F | 2.837 | |
| Adj $R^2$ | 0.0461 | |

Note: ***: $p < 0.01$, **: $p < 0.05$, *: $p < 0.1$.

(2) Robustness test of mediating effect. Compared with the Sobel method and stepwise regression method, the Bootstrap method has become a commonly used method in the testing of mediating effects [91] because it allows more accurate confidence intervals and higher testing power to be obtained [92]. Therefore, this study used this method to conduct a robustness test. The results show that the direct effect of network embeddedness on farmers' participation in IRHS was 0.130, and the P-value was 0.279, which was relatively close to the significance level of 10%. The confidence interval of the indirect effect was (0.007, 0.2715), and the P-value was 0.075. Therefore, the indirect effect was significant. In conclusion, environmental awareness had a mediating effect between network embeddedness and farmers' centralized treatment of domestic sewage thus, hypothesis H3b was verified.

## 6. Discussion and Policy Implications

### 6.1. Discussion of Findings

This paper constructed a theoretical framework for farmers' participation in IRHS based on social embeddedness theory to elaborate the mechanism underlying farmers' environmental behavior and empirically analyzed the results of the theoretical analysis to obtain the following two main conclusions.

On the one hand, relational embeddedness has a significant direct negative impact on participation in the centralized treatment of domestic sewage However, neither network embeddedness nor environmental awareness had a significant effect on farmers' participation in the centralized treatment of domestic waste. In this study, the relational embeddedness analysis consolidated some findings obtained from analysis of the farmers' social networks [64,65,69], and also provided additional quantitative results based on recent research [52]. This study also clarified the formation mechanism underlying farmers' participation in IRHS. Additionally, and surprisingly, compared with the formation mechanism underlying farmers' participation in the centralized treatment of domestic sewage, network embeddedness and environmental awareness could not explain farmers' participation in the centralized treatment of domestic waste. Because the state vigorously promotes the construction of an ecological civilization [93] and has issued many policies and regulations to guide and encourage ecological environmental protection [94,95], the state has invested in waste treatment facilities through rural infrastructure construction (e.g., organizing staff to clean up garbage ponds regularly, and regular inspections of villages by cadres). The implementation of these measures has increased the centralized treatment of rural domestic waste and sewage by farmers. Local administrative organizations, as grass-roots governments, still need to continue strengthening the input of various policy measures to better maintain the treatment of rural domestic waste and to protect the local ecological environment [96].

On the other hand, environmental awareness has a significant promoting effect on participation in the centralized treatment of domestic sewage, and structural embeddedness further influences farmers' centralized treatment of domestic sewage through their environmental awareness. Environmental awareness plays an important mediating role be-

tween structural embeddedness and participation in the centralized treatment of domestic sewage. To a large extent, the results of the current study are consistent with the analysis of environmental behavior [23] and positive environmental protection behaviors [73–77]. The research results also echo the finding that good environmental awareness will ultimately lead to good individual environmental behavior [73]. Furthermore, structural embeddedness can strongly facilitate farmers' centralized treatment of domestic sewage through their environmental awareness. This finding is in line with those of many existing studies [73,75,79,80].

*6.2. Policy Implications*

Based on the above results, this paper suggests the following policy recommendations: Firstly, encourage the establishment of a promotion and discussion network to involve farmers in, embed it in the local villagers' network, and strengthen the guidance of farmers' environmental behavior. As an informal relationship network, the villagers' network has the characteristics of "homogeneity", "locality", and "atomicity". It is suggested that farmers should support exchanges and studies among themselves and to encourage them to integrate themselves into the "heterogeneity" network of rural elites, we will give full play to the role of the rural relationship network (especially the village cadres and rural elites) in promoting their participation in IRHS. Secondly, support education and dissemination of environmental knowledge among rural households and increase especially the supply of environmental knowledge to rural households in hilly areas; through the network, television, and other channels, increase the publicity of rural environmental protection and enhance their responsibility and awareness of environmental protection. Furthermore, increase the overall level of farmers' environmental awareness in China to stimulate more farmers to participate in IRHS. Thirdly, strengthen the relationship between farmers, family members, and village cadres, which can promote participation in domestic sewage centralized treatment. Closer relationships between farmers, family members, and village cadres as well as higher levels of their environmental awareness can progressively promote their participation in IRHS.

**7. Conclusions**

This paper focused on the analysis of farmers' participation in improving rural human settlements (IRHS) from two perspectives: the centralized treatment of domestic waste and domestic sewage. The influence of network embeddedness and environmental awareness on farmers' participation in IRHS was discussed, and an empirical test using the survey data of 495 rural households in Hubei Province was conducted. The research conclusions are as follows: firstly, relational embeddedness can effectively promote farmers' participation in domestic sewage centralized treatment. Because of restrictions deriving from many sources, farmers' relational embeddedness negatively influences their participation in the centralized treatment of domestic sewage. To some extent, this indicates that relationship embedding can effectively promote farmers' participation in the collective action of improving rural human settlements. Secondly, environmental awareness can effectively promote farmers' participation in the centralized treatment of domestic sewage. The promoting effect on the centralized treatment of domestic sewage increases along with the environmental awareness of farmers. To some extent, this indicates that environmental awareness can effectively encourage farmers to participate in the collective action of improving rural human settlements. Finally, structural embeddedness further promotes farmers' participation in the centralized treatment of domestic sewage by improving their environmental awareness. Environmental awareness plays an important mediating role between structural embeddedness and the centralized treatment of domestic sewage. To some extent, this indicates that structural embeddedness can further promote farmers' participation in the collective action of improving rural human settlements through enhancing environmental awareness.

Despite its valid contributions, this study has certain limitations and unanswered questions that require further study. Firstly, although the study area was only located in central China and the random sampling method was adopted to obtain the data to minimize sampling bias, the sample size mainly covered two important typical geomorphic types (e.g., plains and mountains), similar to the geomorphic types in other countries (e.g., the United States and Brazil). Therefore, although the interpretation of the results should be cautious, they can be generalized to other developed and developing countries. The importance of these results may be reinforced by the fact that they varied widely among farmers in different geographical settings. Secondly, exploring the corresponding consequences of farmers' participation in IRHS would be a natural extension of this study. Specifically, future research should investigate the social, economic, and ecological effects of farmers' participation in IRHS. Thirdly, this paper mainly employs social embeddedness theory to analyze the mechanisms influencing farmers' participation in IRHS. Nevertheless, specific types of network embeddedness not only affect farmers' behaviors but also play important roles in the formation of combinations. Since this study did not provide a detailed analysis of these combinations of types of network embeddedness, it is not clear which combinations are most suitable to motivate farmers to participate in IRHS. To address the deficiency of traditional regression analysis, which only analyzes the net effect, future research should explore the mechanisms through which networks influence farmers' participation in IRHS from the perspective of network configuration. Finally, the concept of IRHS is broad. Although based on the actual situation, this paper mainly focuses on the centralized treatment of domestic waste and domestic sewage centralized treatment; other aspects of IRHS can be further explored in future studies.

**Author Contributions:** Conceptualization, J.X.; methodology, J.X.; validation, J.X.; formal analysis, J.X.; data curation, J.X., and W.X.; writing—original draft preparation, J.X.; writing—review and editing, J.X.; supervision, G.Y. and G.W.; funding acquisition, G.W. All authors have read and agreed to the published version of the manuscript.

**Funding:** This research was funded by the National Natural Science Foundation of China (Project Number: 71901101 and 71973050), the Fellowship of China Postdoctoral Science Foundation (Project Number: 2020M671134), and the Fundamental Research Funds for the Central Universities, China (Program Number: 2021JC002).

**Institutional Review Board Statement:** Not applicable.

**Informed Consent Statement:** Not applicable.

**Data Availability Statement:** The data presented in this study are available on request from the corresponding author.

**Acknowledgments:** We would like to thank Kaili Peng and Hongbo Li, Huazhong Agricultural University, for their support of this work.

**Conflicts of Interest:** The authors declare no conflict of interest.

## Notes

[1]  During the process of investigation, it appeared that some farmers may treat centralized and non-centralized domestic waste and domestic sewage. Farmers dealing with domestic waste and sewage might not only carry out centralized treatment but also occasionally dump waste or dirty water. In this study, performing centralized and non-centralized treatment simultaneously was considered as not participating in IRHS. Due to the restrictions of past living habits and other factors, there are still some environmentally unfriendly behaviors among farmers, which cause some damage to the environment. Therefore, this behavior is still regarded as not participating in IRHS.

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
