# Peer review of "How Do Network Embeddedness and Environmental Awareness Affect Farmers’ Participation in Improving Rural Human Settlements?"

_land, doi:10.3390/land10101095_

Round 1
Reviewer 1 Report
1) The study focuses on suburban villages, which are not the same as typical (classical) rural villages.
Please clarify this!
2) Thanks for the very actual topic!
Author Response
We have responded one by one based on your review comments, please see the attachment for details.

Reviewer 2 Report
The paper "How do the network embeddedness and environmental cognition affect farmers’ participation in improving rural human settlement?" is very well written. The scientific part is as well perfectly analyzed.
I recommend the paper for acceptance for publication with no reservations
Author Response
Thank you very much for your comments and suggestions on this paper. Please refer to the attachment for the specific modification details.

Reviewer 3 Report
The paper presents interest and has some good elements, but I am wondering why the authors followed a such complicate path?
Specifically:
1. The authors using the concept of the improve rural human settlement (IRHS) -in the title and in the whole manuscript-. But according the definitions of IRHS which authors present into Introduction section, the paper exams partly this concept, exploring only the improvement of domestic waste centralized treatment and domestic sewage centralized treatment. The IRHS is a wider concept who cover many aspects and in my opinion the authors must disconnect the paper with the IRHS. In the title suggest also to add the country of the study (for example: How do the network embeddedness and environmental cognition affect farmers’ participation in improving of the centralized treatment of domestic waste and domestic sewage into the rural settlements in China).
2. The Introduction section contains appropriate material and data but it is necessary to presents clearer the purpose and the objectives of the research (and in other points of the manuscript too).
3. In the "Theoretical analysis" section, in my opinion, the authors trying to based (partly also) on the social embeddedness theory. In my opinion it is not necessary (and I do not know if it is right). The Figure 1 contains a logical view which lead in relative research hypotheses. In the theoretical analysis I suggest to the authors to erase the first general paragraphs (about social embeddedness theory) and to replace them with a analytical description (based on literature review) about the condition in rural areas and settlements, especially in China. About also the human relationships and the changes in them in the last decades. The bond and the barriers between the people in rural settlements, the policies for rural life improvement, etc.
4. The discussion section needs improvement. The first paragraph, in my opinion is more useful in another point of the manuscript (possible to Introduction or to Theoretical framework). The rest of the section is general and the comparison with similar studies is very weak. I suggest to authors to compare their research results one by one with similar international oriented studies.
5. In the Conclusion section the authors simply repeated the results of the research and present mainly the limitations of research and not the Conclusions. Please, add Conclusions by avoiding repetitions.
6. In the Conclusion section to the limitations paragraph the authors mentioning that: "the sample size mainly included two important typical geomorphic types, similar to other countries (e.g., the United States and Brazil)". Which are these two geomorphic types? and Why the results can be generalized to other developed and developing countries? In my opinion the results cannot generalized because although the sample size was 495 respondents (heads of rural households) we do not know the exact size of the research population.
7. In line 282 the authors mention: "Three or four towns (or town- ships) were randomly selected from these five sample areas ..." How many towns exactly 3 or 4 and were towns or townships?
8. In whole manuscript there are repetitions. Please check and avoid these repetitions because they do not help the readers.
Author Response
Thank you very much for your comments and suggestions on this article. Please refer to the attachment for the specific modification details.

Round 2
Reviewer 3 Report
No comments